# Identification and characterization of a novel heparan sulfate-binding domain in Activin A longest variants and implications for function

Evan Yang[1][¤], Christina Mundy[1], Eric F. Rappaport[2], Maurizio Pacifici[1], Paul C. Billings[1]*

**1** Translational Research Program in Pediatric Orthopaedics, Division of Orthopaedic Surgery, The Children's Hospital of Philadelphia, Philadelphia, Pennsylvania, United States of America, **2** Molecular Genetics Core, The Children's Hospital of Philadelphia, Philadelphia, Pennsylvania, United States of America

¤ Current address: Graduate Program in Biological and Biomedical Sciences (BBS), Harvard Medical School, Boston, Massachusetts, United States of America

* BillingsP@email.chop.edu

**Data Availability Statement:** All relevant data are within the paper and its Supporting Information files.

## Abstract

Activins regulate numerous processes including inflammation and are synthesized as precursors consisting of a long N-terminal pro-region and a mature protein. Genomic human databases currently list three activin A (Act A) variants termed X1, X2 and X3. The X3 variant is the shortest, lacks N-terminal segments present in X1 and X2, and has been the focus of most past literature. Here, we asked whether these variants are expressed by human cells and tissues and what structural features are contained within their pro-regions. Human monocytic-like cells THP1 and U937 expressed X1 and X2 variants after exposure to phorbol ester or granulocyte-macrophage colony-stimulating factor, while X2 transcripts were present in placenta. Expression vectors encoding full length X2 or X3 variants resulted in production and secretion of biologically active Act A from cultured cells. Previous studies reported a putative HS-binding domain (HBD) in the X3 pro-region. Here, we identified a novel HBD with consensus HS-binding motifs near the N-terminal end of X1 and X2 pro-regions. Peptides encompassing this new domain interacted with substrate-bound HS with nanomolar affinity, while peptides from putative X3 HBD did not. In good agreement, full length X2 pro-region interacted with heparin-agarose, while the X3 pro-region did not. In sum, our study reveals that Act A variants are expressed by inflammatory cells and placenta and yield biological activity. The high affinity HBD in X1 and X2 pro-region and its absence in X3 could greatly influence overall Act A distribution, availability and activity in physiological and pathological circumstances.

## Introduction

Activins are members of the transforming growth factor-β (TGF-β) superfamily of signaling and growth factor proteins that include bone morphogenetic proteins (BMPs) and growth and differentiation factors (GDFs) [1, 2]. Activin A (Act A) was originally purified from gonadal

**Funding:** This work was supported by grants RO1AR061758 and RO1AR071946 to MP from the Institute of Arthritis, Musculoskeletal and Skin Disease at the National Institutes of Health.

**Competing interests:** The authors have declared that no competing interests exist.

**Abbreviations:** Act A, activin A; BMP, bone morphogenetic protein; CW, Cardin-Weintraub motif; Est, expressed sequence tag; FOP, Fibrodysplasia Ossificans Progressiva; Fs, follistatin; GM-CSF, granulocyte macrophage-colony stimulating factor; HBD, HS/heparin binding domain; HME, Hereditary Multiple Exostoses; HS, heparan sulfate; MGC, Mammalian Gene Collection; PMA, phorbol 12-Myristate 13-Acetate; TGF-β, transforming growth factor β.

fluids and found to stimulate release of pituitary follicle stimulating hormone [3, 4]. Since then, it has become apparent that Act A regulates many physiologic processes and events including cell proliferation and differentiation, wound repair and immune responses [5–8]. Changes in Act A availability or function contribute to various pathologies [9].

As other TGF-β superfamily members, Act A is initially synthesized as a pre-pro-protein that encompasses a large N-terminal pro-region and a smaller mature protein portion which is subsequently dimerized by a C-terminal disulfide bond [10, 11]. While transitioning through the secretory pathway, the precursors undergo processing by furin-like proteases that cleave them at a prescribed and obligatory cluster of basic amino acids, but do not disrupt the association of pro-region and mature protein [8, 12, 13]. The processed pro-region/mature protein dimeric complexes are finally secreted, and evidence indicates that they can exert autocrine or paracrine roles in multiple tissue and organ functions [8, 9]. Previous studies have examined activin interactions with protein antagonists including follistatin (Fs) [14] and the ability of activins to form diffusion gradients [15], presumably via interactions with extracellular matrix components [16].

The NCBI database lists three human Act A variants, termed X1, X2, and X3. These variants were initially identified and sequenced by the Mammalian Gene Collection (MGC) Program Team as expressed sequence tags (ESTs) [17], but their differences in expression or function have yet to be examined empirically in human cells and tissues. The three variants are predicted to contain the same mature protein, but differ in the length of their pro-region as a result of alternative splicing and translation start site. Much of the past and recent literature has focused on the shortest X3 variant that lacks the longer N-terminal segments characterizing the X1 and X2 variants [8, 11]. In vitro studies have indicated that bioactivity of the entire X3 Act A protein complex is similar to that of the mature protein alone, suggesting that the X3 pro-region may have roles other than influencing ligand availability or receptor-ligand interactions [18]. A previous structural analysis of the X3 variant reported the presence of a putative domain in the middle of the pro-region seemingly able to interact with heparan sulfate (HS) and heparin [19]. To date, similar analyses have not been reported for the X1 and X2 variants.

In other TFG-β superfamily members and BMPs in particular, there is ample evidence that their distribution, availability, bioactivity and turn over are tightly regulated by interactions with HS-rich proteoglycans on the cell surface and extracellular matrix [20, 21]. In pioneering studies, Cardin and Weintraub were among the first investigators to identify and characterize protein domain(s) responsible for interaction with HS [22]. Comparative analyses on vitronectin, platelet factor-4, apoliprotein A and apoliprotein B led them to identify two HS-binding amino acid sequences -XBBXBX and XBBBXXBX-, thus named CW motifs, where B represents basic residues (usually Arg or Lys) interacting with reciprocally-charged sulfated carbohydrate residues and X represents non-charged residues. Subsequent studies by multiple investigators have verified these original findings and identified similar motifs or variations in many other proteins [23, 24]. Notably, the HS chains themselves are endowed with remarkable variability in their patterns of: O-sulfation at C2, C3 and/or C6 positions and N-deacetylation and N-sulfation in glucosamine residues, and C5 epimerization of D-glucuronic acid to L-iduronic acid residues [25, 26]. These overall modifications create short segments with specific sulfation patterns and sugar structure that exhibit binding avidity toward different CW domains on various proteins [27].

One example of the fundamental importance of HS-protein interactions was provided by studies showing that mutation of basic residues within the HS-binding domain (HBD) of BMP2 and BMP4 reduced their ability to interact with HS, but greatly increased their biological activity and potency as revealed by in vitro and in vivo assays [28, 29]. These studies showed that at least for certain BMPs, interactions with HS normally serve to restrain, delimit

and define their biological activities [30]. Further, we recently showed that while the HBD in BMP2 and BMP4 resides near the N-terminus of the mature protein, that domain resides at the C-terminus in BMP5, BMP6 and BMP7 [31]. Structural analyses suggested that the HBDs in these two BMP subgroups have distinct structural characteristics and spatial orientation, indicating that HBD location and structure are protein-specific [32]. In good agreement, we also found that the congenital HS deficiency characteristic of the pediatric disorder Hereditary Multiple Exostoses (HME) triggers excessive and ectopic canonical BMP signaling in growth plate perichondrium and leads to exostosis (osteochondroma) formation [33, 34], demonstrating that aberrations in normal HS-protein interactions can have severe pathological consequences.

In the present study, we have carried out a detailed examination of Act A variants and in particular, asked whether the X1 and X2 variants are expressed by human cells and tissues and what possible additional structural features may be provided by their longer N-terminal pro-regions. Our data reveal for the first time that the X1 and X2 variants are in fact expressed by human cells and placenta and contain a previously unidentified high affinity HBD toward the N-terminal end of their pro-region that is absent in the X3 pro-region. The diverse structural features of the variants could influence overall Act A availability, range and activity in physiological and pathological circumstances.

## Results

### Human inhibin genes and expression variants

Previous studies have provided general information and insights into the genomic organization of the human inhibinβ genes, designated *INHβA*, *INHβB*, *INHβC* and *INHβE*, that encode Act A, activin B (Act B), activin C (Act C) and activin E (Act E), respectively [35–37]. To gain further structural details, we relied on the UCSC Genome Browser (genome.ucsc.edu) and NCBI sequence and gene structure databases. Computer-driven algorithms indicated that human INHβA resides on the short arm of chromosome 7 and is characterized by 7 coding exons spanning an overall genomic segment of about 25Kb (Fig 1, Tables 1–3 and S1 Fig).

The NCBI database lists three variants for human INHβA designated X1, X2 and X3 and alignment of the nucleotide and amino acid sequences indicate that the variants are the product of alternative splicing and translation start sites (Fig 1). The X1 variant is encoded by exons 2, 3, 6, and 7 and the X2 variant is encoded by exons 1, 6, and 7. The X3 variant includes exons 6 and 7 only, but results from the use of an alternative translation start site within exon 6 (Fig 1, arrow in X3). The mature protein is encoded by approximately the 3' half of exon 7 and is common to all Act A variants, which differ in the length of their pro-regions (Fig 1 and S2 Fig). It remains to be clarified whether exons 4 and 5 are incorporated into other as yet undiscovered Act A variants, if other potential exons reside upstream of coding exon 1, and what are the promoter/enhancer elements regulating *INHβA* gene expression [1].

It should be noted that the above transcript/protein predictions of variants are based on computational genomic analyses that were conducted initially by the MGC team using ESTs to explore the transcriptome and then subsequently confirmed by whole genome sequencing and bioinformatics [17]. Much of the past and recent extensive literature on Act A has utilized recombinant forms of the X3 variant [14, 18], but currently there is little information about the expression and structural characteristics of the X1 and X2 variants. Thus, we first asked whether these variants are expressed by myeloid- lineage cells that are amongst the cell types known to produce Act A [38, 39]. For these studies, we utilized the human monocytic-like cell lines THP1 and U937 that have the capacity to differentiate into macrophages following appropriate stimulation [40]. For initial experiments, cells were treated with phorbol 12-Myristate

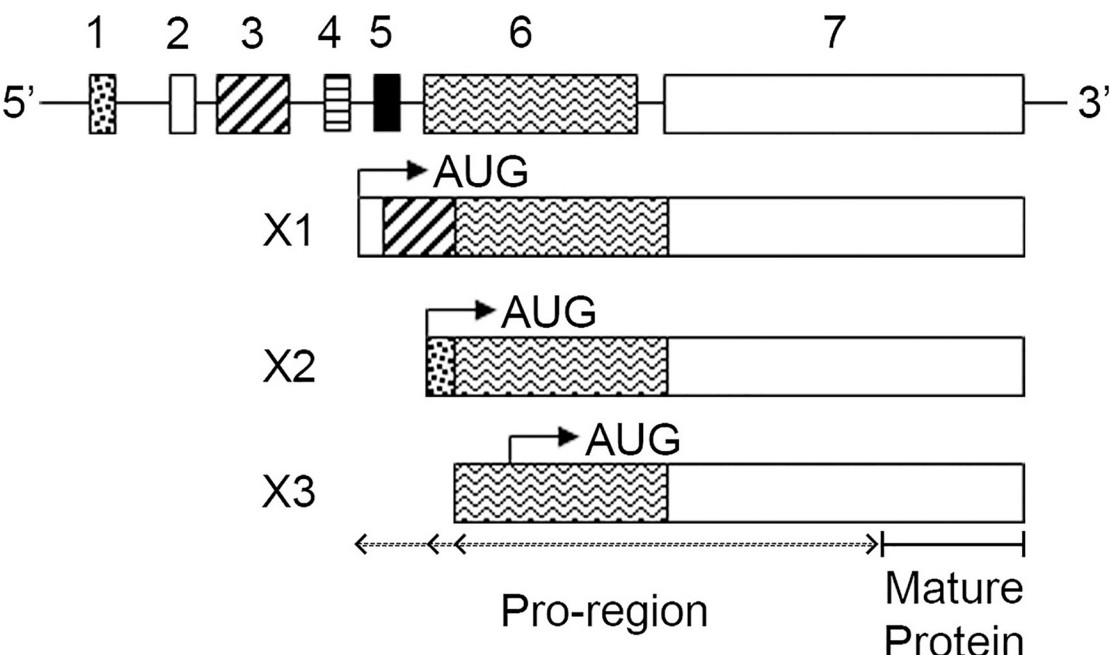

**Fig 1. Human *INHβA* genomic organization.** Human *INHβA* is located on Chr 7 and includes seven coding exons. Arrows indicate AUG translation start site in each variant. The X1 variant is encoded by exons 2, 3, 6, 7, the X2 variant is encoded by exons 1, 6 and 7, and the X3 variant is encoded only by exons 6 and 7 but utilizes an alternative AUG start site within exon 6. This information was obtained from the UCSC Genome Browser (genome.ucsc.edu) and NCBI GenBank. Lines connecting each exon denote introns and are not drawn to scale. Also see Tables 1–3.

13-Acetate (PMA) for 4 hrs to induce differentiation, total cellular RNAs were isolated and processed for cDNA synthesis and semi-quantitative PCR. We used unique forward primers anchored at the ATG start sites of X1 or X2 variants and combined with a common reverse primer that includes the TAG stop codon in exon 7 (Table 4). We found that PMA stimulated expression of transcripts encoding the X2 variant in U937 cells, whose identity was confirmed by Sanger sequencing of the purified PCR product (Fig 2A, lane 2, S5 Fig); however, expression of the X1 variant was undetectable (Fig 2A, lane 8). Because the X3 variant shares its entire composition with the X1 and X2 variants and does not have a unique 5' end (Fig 1), we did not test its expression using this approach. Granulocyte macrophage-colony stimulating factor (GM-CSF) induces strong expression of Act A in purified human monocytes [41]. Thus, U937 and THP1 cells were untreated or treated with GM-CSF (1 ng/ml; 8 hr) and the resulting RNA

**Table 1. Chromosomal location and genomic organization of human activin genes.**

| Gene | Location[1] | Gene ID[2] | Exons | Start/Stop[3] | Size[4] |
|------|-----------|-----------|-------|---------------|---------|
| INHβA | 7p14.1 | 3624 | 7 | chr7: 41,685,101–41,710,532 | 25.4 |
| INHβB | 2q14.2 | 3625 | 2 | chr2: 120,346,142–120,351,808 | 5.7 |
| INHβC | 12q13.3 | 3626 | 2 | chr12: 57,434,685–57,452,062 | 17.4 |
| INHβE | 12q13.3 | 83729 | 2 | chr12: 57,455,291–57,458,013 | 2.7 |

[1]. Chromosomal location determined using UCSC Genome Browser

[2]. NCBI Gene Identification

[3]. Region on chromosome

[4]. Size of gene in kb.

**Table 2. Exon locations within human activin genes.**

| Activin[1] | Location[2] |
|---|---|
| INHβA (3624) | Chr 7:41685101–41710532 |
| Exon 1 | Chr 7:41708853–41710532 |
| Exon 2 | Chr 7:41706187–41706314 |
| Exon 3 | Chr 7:41705284–41705471 |
| Exon 4 | Chr 7:41704729–41704784 |
| Exon 5 | Chr 7:41703005–41703108 |
| Exon 6 | Chr 7:41699987–41700517 |
| Exon 7 | Chr 7:41685101–41690542 |
| INHβB (3625) | Chr 2:120346142–120351808 |
| Exon 1 | Chr 2:120346142–120346636 |
| Exon 2 | Chr 2:120349099–120351808 |
| INHβC (3626) | Chr 12:57434685–57452062 |
| Exon 1 | Chr 12:57434685–57435199 |
| Exon 2 | Chr 12:57449277–57452062 |
| INHβE (83729) | Chr 12:57455291–57458013 |
| Exon 1 | Chr 12:57455291–57455834 |
| Exon 2 | Chr 12:57456094–57458013 |

[1]. Activin genes. Number in parenthesis: NCBI Gene identifier

[2]. Position on indicated chromosome was determined using the BLAT search engine, UCSC Genome Browser, using the Human Dec. 2013 (GRCh38/hg38) Assembly (genome.ucsc.edu).

samples were processed for PCR analysis as above. Interestingly, GM-CSF upregulated the X2 variant in both cell lines (Fig 2A, Lanes 3, 6), but only stimulated expression of the X1 variant in U937 cells (Fig 2A, lane 9). To make sure both cell lines have the cell surface machinery to respond to GM-CSF, we subjected them to FACS analysis using antibodies to its receptor CD116 or pre-immune IgGs as control. Both cell lines exhibited readily detectable levels of CD116 (Fig 2B and 2C).

To determine which Act A variant(s) are expressed under normal physiological conditions, we prepared cDNA from human placental RNA and processed it for PCR using different primer sets. A first set amplified an amplicon covering exons 6 and 7, present in all three Act A variants (Fig 3A and 3C, black arrows). Next, we used primer sets containing unique forward primers specific for the 5' ends of X1 or X2 variants (Fig 3B, white arrows). This yielded a

**Table 3. Human activin isoforms and variants.**

| Isoform | Accession[1] | Accession[2] | Size[3] |
|---|---|---|---|
| INHBA X1 | XM_017012174 | XP_016867663 | 548 (116) |
| INHBA X2 | XM_017012175 | XP_016867664 | 483 (116) |
| INHBA X3 | XM_017012176 | NP_002183 | 426 (116) |
| INHBB | NM_002193 | NP_002184 | 407 (115) |
| INHBC | NM_005538 | NP_005529 | 352 (116) |
| INHBE | NM_031479 | NP_113667 | 350 (114) |

[1]. Nucleotide Accession number

[2]. Protein Accession number

[3]. Amino acids in full-length protein (number of residues in mature ligand).

**Table 4. PCR Primers used to amplify human activin transcripts.**

| Primer | Gene | Accession[1] | Sequence (5'-3') | Position[2] |
|---|---|---|---|---|
| 1 | INHβA (X1) | XM_017012174 | F: ATGCCAACTTTGAACAGGA | 1774–1791 (Start) |
| 2 | INHβA (X2) | XM_017012175 | F: ATGGAACTTATTACCCAAGGG | 1653–1673 (Start) |
| 3 | INHβA | XM_017012175 | F: AATTTGCTGAAGAGGAGAAG | 1683–1702 (Ex 6) |
| 3 | INHβA | XM_017012174 | R: TGACTCGGCAAACGTGATGATCTCCG | 2526–2501 |
| 4 | INHβA | XM_017012175 | R: CTATGAGCACCCACACTC | 3104–3087 (Stop) |
| 5 | INHβB | NM_002193 | F: ATGGACGGGCTGCCCGGTCG | 48–67 (start) |
| 6 | INHβB | NM_002193 | R: ACCCTCTTCTCCACCATGTTC | 703–683 |
| 7 | INHβB | NM_002193 | F: AAGGTGCGGGTCAAAGTGT | 633–651 |
| 8 | INHβB | NM_002193 | R: CCGTGCCCCTGCCTTGCACT | 1292–1273 (Stop) |
| 9 | INHβC | NM_005538 | F: ATGACCTCCTCATTGCTTC | 203–221 (Start) |
| 10 | INHβC | NM_005538 | R: CTAACTGCACCCACAGG | 1261–1245 (Stop) |
| 11 | INHβC | NM_005538 | F: CACTTCTCCTCTGATAGAACTG | 551–572 |
| 12 | INHβC | NM_005538 | R: TGCCACAAAAGGCCTAT | 868–852 |
| 13 | INHβE | NM_031479 | F: ATGCGGCTCCCTGAT | 247–261 (Start) |
| 14 | INHβE | NM_031479 | R: CTAGCTGCAGCCACAG | 1299–1284 (Stop) |
| 15 | INHβE | NM_031479 | F: CTCCCTG CTCACTTTTC | 564–580 |
| 16 | INHβE | NM_031479 | R: AGTCTAGTTGCAGTTTCAG | 820–802 |

[1]. NCBI Accession number

[2]. Annealing position of primer

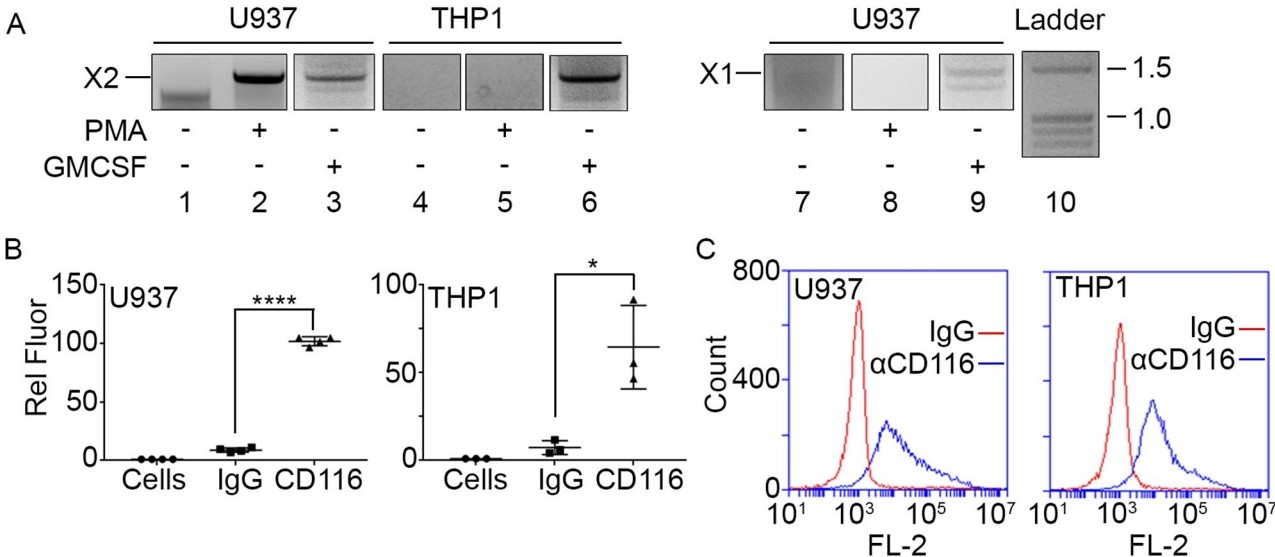

**Fig 2. Human monocytic-like cells express Act A X1 and/or X2 variants and CD116.** *A*, **Cells were treated** with PMA or GM-CSF. Following treatment, RNAs were processed for PCR analysis of Act A variant expression. Amplified bands were purified from the gels and subjected to Sanger sequencing to confirm transcript identity. Treatment of U937 cells with PMA induced expression of Act A-X2 (lane 2), while treatment with GM-CSF induced expression of both Act A-X2 and -X1 (lanes 3, 9). Treatment of THP-1 cells with GM-CSF induced expression of Act A-X2 (lane 6). Lane 10, Fisher exACTGene DNA ladder. *B* and *C*, Scatter plot and histogram depicting CD116 detection. U937 and THP1 cells were incubated with anti-CD116-PE antibody or IgG-PE control. The cells were washed, and antibody binding was determined by FACS. Note that both cell lines contain cell surface CD116. Statistical Analysis (Student's t test) for B: IgG vs CD116: ****: $P < = 0.0001$, *: $P < = 0.05$.

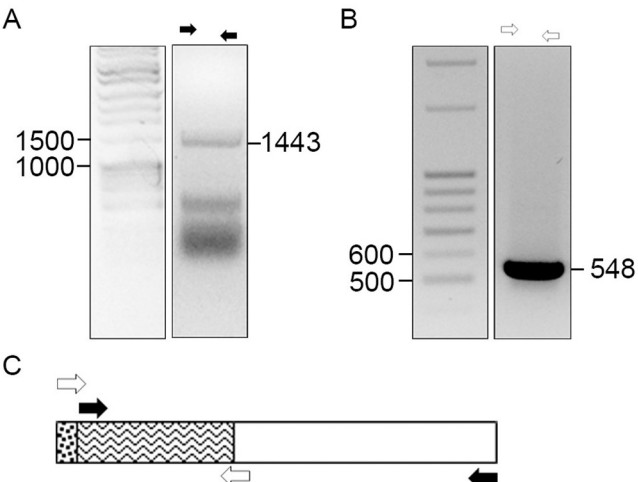

**Fig 3. Normal human placenta tissues contain transcripts encoding Act A X2 variant.** cDNA was prepared from human placenta RNA and used as template to amplify Act A transcripts by PCR. The PCR products were size fractionated on 1% agarose gels, and bands were purified and subjected to Sanger sequencing to confirm transcript identity. *A*, amplification of exons 6 and 7 using primers indicated by black arrows, yielded a predicted product of 1,443 bp. *B*, amplification of 5' end of Act A-X2, using primers indicated by clear arrows, yielded a predicted amplicon of 548 bp. *C*, schematic representation of X2 variant and primer sets (White and Black arrows) used to amplify products shown in *A* and *B*. Size markers; Fisher exACTGene DNA ladder is on the left, lane 10. Also see Fig 1.

product encoding the 5' region of the X2 variant, but no amplification product homologous to the X1 5' region was obtained (not shown). Thus, the Act A X2 variant is expressed in human placenta.

Together, the data provide new evidence that Act A X1 and X2 variants are expressed by differentiating human monocytic cell lines and human tissue. Because the X1 and/or X2 variants are expressed in response to specific chemical or biological cues, it is possible that they may be expressed and utilized in vivo during specific stages of development and under certain pathologic conditions.

## AD293 and COS1 cells secrete biologically active Act A following transfection with X2 and X3 expression vectors

Next, we explored the biological activity of Act A produced and secreted by cultured cells. The entire open reading frame encoding each variant was cloned into the mammalian expression vector pcDNA3.1+, which utilizes the CMV promotor to drive protein expression. Following verification of constructs by DNA sequencing, each expression construct along with a construct expressing green fluorescent protein (GFP) was transfected into AD293 cells using FuGENE 6 Transfection Reagent (Promega), resulting in a transfection efficiency of > 90% (Fig 4B). Three days after transfection, conditioned medium was harvested and the remaining cells were scraped from the dishes, RNA was isolated, treated with DNAse and subsequently used for cDNA synthesis. Cells transfected with each Act A variant expressed the appropriate mRNA (Fig 4C, lanes 3–5). Removal of plasmid DNA was confirmed using PCR primers specific for beta-lactamase, which encodes the AmpR gene on pcDNA3.1 expression plasmid (Fig 4C, right). This amplicon was readily amplified from plasmid DNA (Fig 4C, lane 6), but was absent in cDNAs prepared from cells (Fig 4C, lanes 7–10).

To assess protein expression, the conditioned media sample from each transfected cell population was incubated with Talon resin, and the bound proteins were eluted with 0.5M

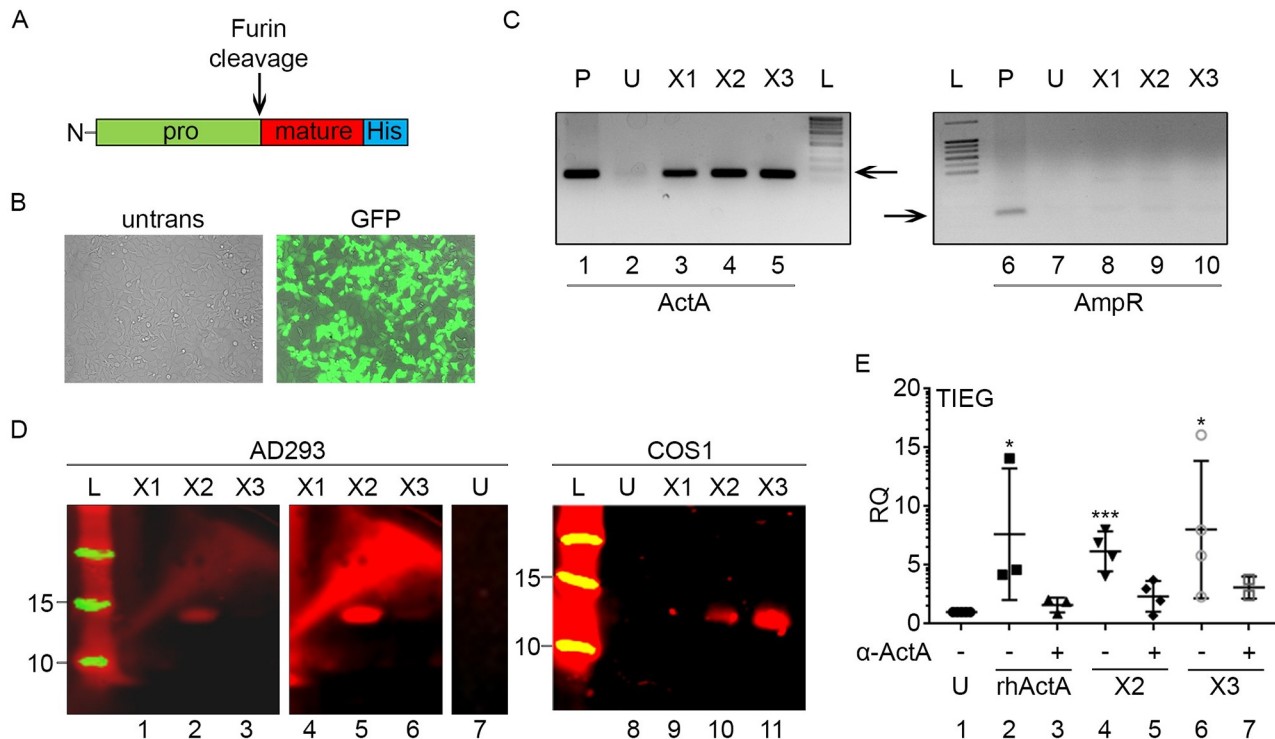

**Fig 4. Expression and secretion of Act A variants.** Expression constructs for full length X1, X2 and X3 variants were cloned into pcDNA 3.1 and transfected into AD293 and COS1 cells. *A*, diagram of Act A expression cassette with His tag on the 3' end. *B*, *GFP* detection indicates that transfection efficiency was > 90%. *C*, RT-PCR analysis (left panel) demonstrating that X1, X2 and X3 transcripts were efficiently expressed following transfection (lanes 3–5), producing an amplicon identical to that of the expression plasmid (lane 1). Untransfected cells are shown in lane 2. RT-PCR analysis (right panel) indicating that plasmid DNA was efficiently removed from our RNA preps (lanes 8–10) but was amplified from plasmid DNA itself (lane 6). D, immunoblots of conditioned medium samples separated under reducing conditions and probed with anti-His tag antibody. Note that samples from AD293 and COS1 cells transfected with X2 or X3 expression constructs exhibit an immunoreactive band ~13 kDa that has the predicted size of mature Act A monomer (lanes 2, 3, 5, 6, 10 and 11). The band was undetectable in conditioned medium from cells transfected with the X1 vector (lanes 1, 4 and 9) or un-transfected cells (lanes 7 and 8). Lanes 4–6 are a longer exposure of lanes 1–3. *E*, THP1 cells were incubated for 2 hrs with conditioned media from AD293 cells that were untransfected (U) or transfected with X2 or X3 constructs. The cells were processed for q-PCR analysis of *TIEG* expression, an early response gene induced by Act A and TGF-β. Note that TIEG expression is up-regulated by treatment with recombinant human Act A (rhActA; lane 2, positive control) as well as conditioned medium from X2 or X3 expressing cells (lanes 4 and 6). This induction was blocked by pre-incubation with a neutralizing antibody to Act A (lanes 3, 5 and 7). Conditioned medium from untransfected cells elicited no TIEG expression (U, lane 1). Statistical Analysis (Student's t test): 1 vs 2, 1 vs 4 and 1 vs 6; *: $P < = 0.05$, ***: $P < = 0.001$.

Imidazole and analyzed on immunoblots. AD293 Cells transfected with X2 or X3 expression vectors both secreted a protein of about 13 kDa, consistent with the mass of mature Act A protein under reducing conditions (Fig 4D, lanes 2, 3, 5 and 6; note that lanes 4–6 are over-exposed images of lanes 1–3). This protein band was more prominent in X2 versus X3 transfected cells, but was not detectable in cells transfected with X1 construct (Fig 4D, lanes 1 and 4). To verify these results, Cos1 cells were transfected with each Act A variant expression construct. Again, immunoblot analysis of conditioned media samples showed secretion of mature Act A ligand only by cells transfected with X2 or X3 variant constructs (Fig 4D, lanes 10 and 11). To test whether the secreted proteins were biologically active, THP1 cells were treated with commercially available recombinant Act A (rhAct A) or conditioned medium samples obtained from cells transfected with the Act A expression constructs. After 2 hr treatment, RNA was extracted, cDNA was prepared and expression of TIEG -an early response gene upregulated by Act A or TGF-β was assessed [42]. As expected, TIEG was rapidly upregulated when cells were treated with rhAct A used as positive control (Fig 4E, lane 2). A similar strong

up-regulation was observed when the cells were exposed to the conditioned medium from cells transfected with X2 and X3 expression constructs (Fig 4E, lanes 4 and 6). Medium from un-transfected cells produced no response (Fig 4E, lane 1). TIEG induction was blocked when the cells were co-treated with a monoclonal Act A neutralizing antibody (Fig 4E, lanes 3, 5 and 7). Thus, AD293 and Cos1 cells transfected with X2 and X3 variant expression vectors secrete biologically active Act A.

## Act A X1 and X2 variants contain a previously unsuspected HS-binding domain (HBD)

Many TFG-β superfamily members are known to establish high affinity interactions with HS-rich proteoglycans on the cell surface and extracellular matrix that regulate protein distribution, availability, bioactivity and turn over [20, 21]. A study published in 2010 analyzed the HS-binding properties of Act A X3 variant and reported the identification of a putative HBD located toward the middle of the pro-region and encompassing amino acids 259 to 272 (Fig 5A) [19]. This segment corresponds to amino acids 381–394 and 316–329 in the X1 and X2 variants, respectively (Fig 5A) and contains several clusters of lysine residues (Fig 5A–5D, highlighted in green). Notably, this domain also contains four glutamic acid residues, which are predicted to interfere with interactions with the anionic sulfate groups in HS. Thus, we closely reexamined the entire amino acid sequence of the X3 variant and compared it to the X1 and X2 variants. We noted the presence of another basic residue-rich domain located near the N-terminal end of the X1 and X2 pro-regions that closely resembled a CW motif (Fig 5A, highlighted in light blue). The domain spans amino acids 78–102 in X1 and amino acids 13–37 in X2, but is absent in X3 (Fig 5A–5D).

To determine whether the newly identified domain is in fact endowed with HS binding capacity, we prepared a synthetic peptide encompassing this region, which was biotinylated on the N-terminus. The domain is identical in X1 and X2 variants and denoted as X2$^{13-37}$ (Fig 6A). In addition, we prepared a biotinylated peptide encompassing the previously described putative HS-binding region in X3 pro-region [19], which is designated X3$^{259-272}$ (Fig 6B). The peptides were tested in solid-phase binding assays, employing 96 well plates pre-coated with HS, following standard protocols established in our previous studies [31]. We found that the X2$^{13-37}$ peptide interacted with HS with high affinity, exhibiting a Kd of 4 ± 2 nM (n = 3); binding was blocked by pre-incubation of the peptide with excess heparin serving as competitor (Fig 6C). As we anticipated given its amino acid composition, the X3$^{259-272}$ failed to interact with HS at any concentration tested (Fig 6C). Changing incubation conditions or lengthening assay time did not change outcomes.

Solid-phase assays are widely used to test macromolecular interactions, but may not fully reflect biological conditions. Thus, we tested the peptides for interaction with the cell surface. U937 cells were lightly fixed with formalin to preserve the cell surface and then incubated with biotinylated peptides complexed with NA-488 [31]. After careful rinsing, cells were processed for FACS analysis. The X2$^{13-37}$ peptide readily and efficiently interacted with the cell surface, and binding was blocked by pre-treatment with heparin (Fig 7A and 7B). However, X3$^{259-272}$ peptide failed to interact with the cells (Fig 7A and 7C), consistent with results from the solid phase binding assay (Fig 6C). To verify assay reliability, we concurrently tested a peptide that encompasses the HBD of BMP4, characterized in a previous study [31]. As expected, the BMP4 peptide effectively interacted with the cell surface and was competed out by excess heparin (Fig 7A). No fluorescence signal was generated by untreated control cells or by cells incubated with NA-488 in the absence of peptides.

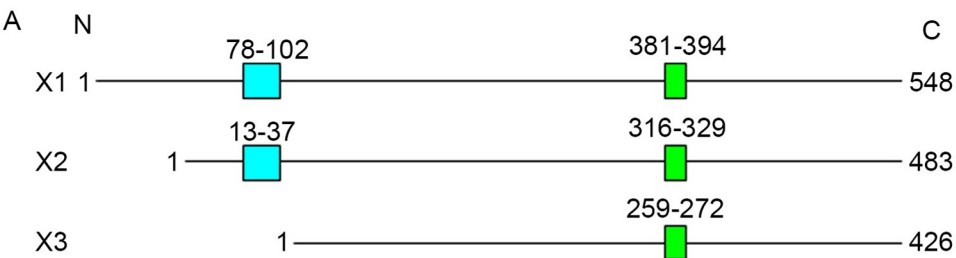

B
**X1**
MPTLNRDSCCHSCQKGGKNQELLLKEVALAGAQGKNRGCRLDRLGPGSRHPASWARAAAAFR
RDPWKLPAGAAQENLL<mark>KRRRKKKTPKKKIKKSTHTKKPAR</mark>EGGGKAGPFKKAITTTFAARM
PLLWLRGFLLASCWIIVRSSPTPGSEGHSAAPDCPSCALAALPKDVPNSQPEMVEAVKKHILNML
HLKKRPDVTQPVPKAALLNAIRKLHVGKVGENGYVEIEDDIGRRAEMNELMEQTSEIITFAESGT
ARKTLHFEISKEGSDLSVVERAEVWLFLKVPKANRTRTKVTIRLFQQQKHPQGSLDTGEEAEEVG
LKGERSELLLSEKVVDARKSTWHVFPVSSSIQRLLDQGKSSLDVRIACEQCQESGASLVLLG<mark>KKK</mark>
<mark>KKEEEGEGKKK</mark>GGGEGGAGADEEKEQSHRPFLMLQARQSEDHPHRRRRR<mark>GLECDGKVNICCKK</mark>
<mark>QFFVSFKDIGWNDWIIAPSGYHANYCEGECPSHIAGTSGSSLSFHSTVINHYRMRGHSPFANLKSC</mark>
<mark>CVPTKLRPMSMLYYDDGQNIIKKDIQNMIVEECGCS</mark>

C
**X2**
MELITQGDPENLL<mark>KRRRKKKTPKKKIKKSTHTKKPAR</mark>EGGGKAGPFKKAITTTFAARMPLLW
LRGFLLASCWIIVRSSPTPGSEGHSAAPDCPSCALAALPKDVPNSQPEMVEAVKKHILNMLHLKK
RPDVTQPVPKAALLNAIRKLHVGKVGENGYVEIEDDIGRRAEMNELMEQTSEIITFAESGTARKT
LHFEISKEGSDLSVVERAEVWLFLKVPKANRTRTKVTIRLFQQQKHPQGSLDTGEEAEEVGLKGE
RSELLLSEKVVDARKSTWHVFPVSSSIQRLLDQGKSSLDVRIACEQCQESGASLVLLG<mark>KKKKKEE</mark>
<mark>EGEGKKK</mark>GGGEGGAGADEEKEQSHRPFLMLQARQSEDHPHRRRRR<mark>GLECDGKVNICCKKQFFV</mark>
<mark>SFKDIGWNDWIIAPSGYHANYCEGECPSHIAGTSGSSLSFHSTVINHYRMRGHSPFANLKSCCVPT</mark>
<mark>KLRPMSMLYYDDGQNIIKKDIQNMIVEECGCS</mark>

D
**X3**
MPLLWLRGFLLASCWIIVRSSPTPGSEGHSAAPDCPSCALAALPKDVPNSQPEMVEAVKKHILNM
LHLKKRPDVTQPVPKAALLNAIRKLHVGKVGENGYVEIEDDIGRRAEMNELMEQTSEIITFAESG
TARKTLHFEISKEGSDLSVVERAEVWLFLKVPKANRTRTKVTIRLFQQQKHPQGSLDTGEEAEEV
GLKGERSELLLSEKVVDARKSTWHVFPVSSSIQRLLDQGKSSLDVRIACEQCQESGASLVLLG<mark>KK</mark>
<mark>KKKKEEEGEGKKK</mark>GGGEGGAGADEEKEQSHRPFLMLQARQSEDHPHRRRRR<mark>GLECDGKVNICCK</mark>
<mark>KQFFVSFKDIGWNDWIIAPSGYHANYCEGECPSHIAGTSGSSLSFHSTVINHYRMRGHSPFANLKS</mark>
<mark>CCVPTKLRPMSMLYYDDGQNIIKKDIQNMIVEECGCS</mark>

**Fig 5. Schematic representation of human Act A variants.** *A*, the three variants, X1, X2, and X3 listed in the NCBI protein database are depicted as a linear segment proportional to their protein sizes that are 548, 483 and 426 amino acids long, respectively. The box colored in aqua depicts the location of the novel high affinity HBD identified in the X1 and X2 variants. The green box depicts the location of the previously-reported region in the X3 variant (presumed to bind to HS) that is also present in the X1 and X2 variants. *B-D*, the entire amino acid sequences of the X1, X2 and X3 variants are shown in which the highlights in aqua and green depict the protein segments as in *A*, while the highlight in yellow depicts the mature protein portion shared by all variants.

To further verify the results, we prepared His-tagged expression constructs in pcDNA3.1 vector encompassing the entire pro-region of X2 and X3 variants and transfected each into AD293 cells. Each construct was efficiently transcribed into mRNA (Fig 8B). Conditioned media samples were incubated with Talon resin or heparin agarose, bound protein was eluted and analyzed on immunoblots with His-tag antibodies. Both pro-regions were produced and secreted by the cells (Fig 8C, lanes 2 and 3). In good agreement with the data above, the secreted X2 pro-region interacted with heparin agarose, while the X3 pro-region did not (Fig 8C, lanes 5 and 6).

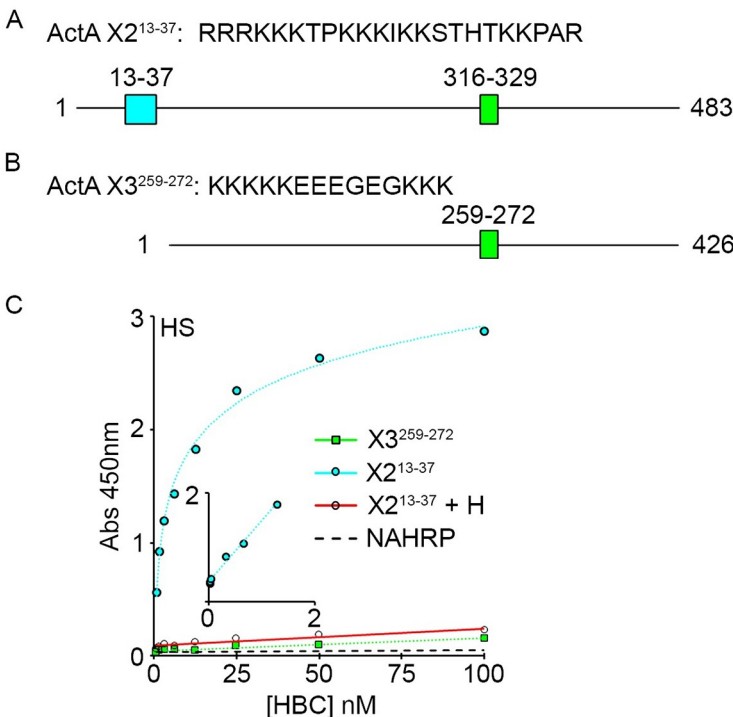

**Fig 6. Peptides corresponding to the novel HBD bind HS with high affinity.** 96-well plates were coated with HS and incubated with tetrameric peptide complexes (HBC) for 2 hrs. Plates were washed, and bound peptides were assessed by addition of OPD substrate. *A and B*, schematic diagram depict the amino acid sequences of Act A- X2[13-37] and Act A- X3[259-272] peptides and their location within the full-length proteins, respectively. *C*, solid phase binding assays of Act A peptides to HS. Note that Act A- X2[13-37] binds HS with high affinity and its binding is blocked by addition of soluble heparin. In contrast, Act A-X3[259-272] displays minimal binding. Inset, double reciprocal plot that yields a Kd of 4 nM (n = 3) for Act A- X2[13-37]. NA-HRP backbone displays minimal binding to HS.

## 3D peptide and protein conformation analyses

The above data predict that the X2[13-37] HBD should possess a 3D configuration and amino acid distribution able to create a surface(s) amenable to interactions with HS. However, these features and properties should be absent or disrupted in the segment encompassing the X3[259-272] segment. To test these predictions, we utilized the I-TASSER server [43] as in previous structural studies on BMP family members [20, 31]. We found that the X2[13-37] domain displayed a helical structure Fig 9A). When examined using helical wheel diagrams, it displayed basic residues (i.e., Lys) mostly on one side, thus generating a cationic surface on one face of the helix, conducive to HS binding (Fig 9A, 9C and 9E). The X3[259-272] segment also displayed a helical configuration, but structural views and helical wheel projections showed that its residues did not elicit a cationic surface suitable for HS interaction (Fig 9B, 9D and 9F). Again, the presence of negative charges from the 4 glutamic acid residues within this sequence are predicted to interfere with HS binding (Fig 9F, highlighted in red).

Next, we used the I-TASSER server to visualize the relative location, orientation and configuration of the X2[13-37] and X3[259-272] domains within the full-length X2 variant. The generated structures revealed that the domains occupy locations on opposite sides of the protein (Fig 10 and S1 and S2 Videos). As also seen in previous crystallography studies of the full-length X3 variant [18], the pro-region is physically separated from the mature protein and connected to it via the latency lasso (Fig 10B). To more clearly determine whether the X2[13-37] domain created an exposed HS-binding surface, we analyzed the structures at different rotational angles.

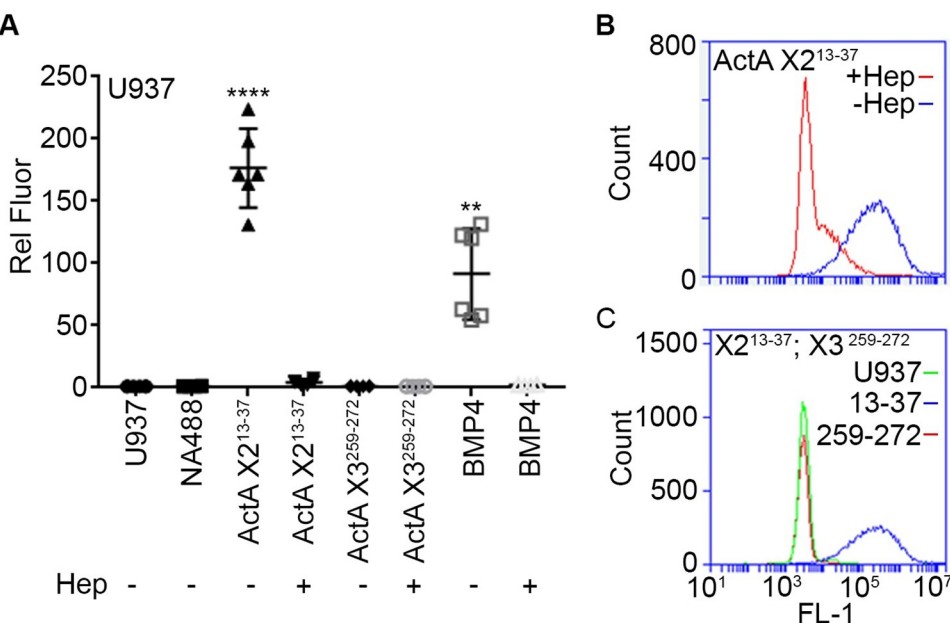

**Fig 7. HBD peptides bind to the cell surface with avidity.** *A*, U937 cells were incubated with peptide complexes consisting of biotinylated peptides pre-incubated with NA-488. Following incubation, the cells were washed, and bound peptide was assessed by FACS analysis. Note that Act A- X2$^{13-37}$ readily binds to the cell surface, and its interaction is blocked by soluble heparin competitor. In contrast, Act A-X3$^{259-272}$ peptide fails to bind. Binding specificity was verified using a previously characterized peptide encompassing the high affinity HBD from BMP4. *B*, representative FACS plots show binding of Act A- X2$^{13-37}$ in the absence (blue line) or presence (red line) of heparin competitor. *C*, representative FACS plots showing binding of Act A- X2$^{13-37}$ (blue line), lack of binding by Act A-X3$^{259-272}$ (red line) and background signal generated by U937 cells alone (green line). Statistical Analysis (Student's t test): NA488 vs ActA X2$^{13-37}$, NA488 vs BMP4: ****: P< = 0.0001, **: P< = 0.01.

Orthogonal views revealed that the domain created a cationic surface facing away from the protein moiety and likely accessible for interactions with HS (see S6 and S7 V\. In contrast, the X3$^{259-272}$ segment was also positioned on the outer perimeter of the protein, but its acidic amino acid composition prevented it from generating a suitable cationic surface (Figs 9 and 10).

## Discussion

Our data reveal that the X1 and X2 variants of Act A are expressed by human inflammatory cells in response to PMA and GM-CSF and in human placenta. The variants differ from each other at the N-terminal end of their pro-region, but share the bulk of the pro-region and the entire mature protein. Our data also show for the first time that the X1 and X2 variants possess a domain in the pro-region that exhibits typical CW characteristics and binds to HS and the cell surface readily and with high affinity. As suggested by 3D stereological analysis, the basic residues in the domain are organized to produce an anionic surface that is adept at HS binding and oriented toward the outer perimeter of the pro-region. Because Act A is assembled and secreted as a dimer held together by an intermolecular disulfide bond [18], each dimer would possess two such HBDs oriented away from each other and potentially functioning autonomously. This is reminiscent of the situation in certain BMP family members such as BMP2 and BMP4 in which the two HBDs in each dimer are distinct in terms of location, action and interactions and act in a bivalent manner in binding assays [29]. However, this appears not be the case for BMP5, BMP6 and BMP7 where the two HBDs come together to form a single

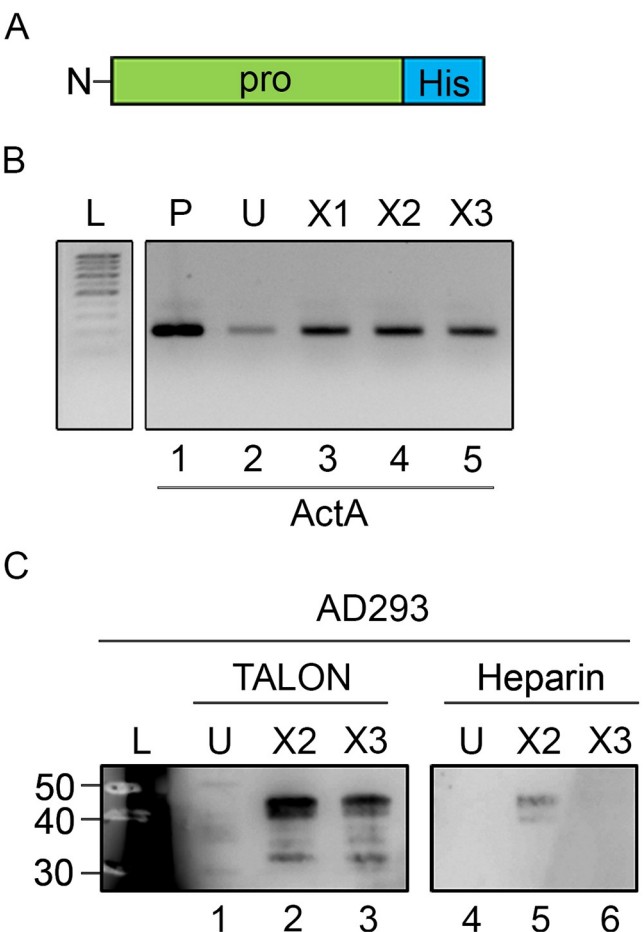

**Fig 8. Expression and binding characteristics of Act A X2 and X3 pro-regions.** *A*, diagram of Act A pro-region expression cassette with HIS tag on 3' end. Cassette was cloned into pcDNA 3.1. Expression constructs encoding the entire pro-region of Act A X1, X2 or X3 variants were transfected into AD293. Cells were processed for RT-PCR analysis four days later. *B*, RT-PCR product analysis shows that each variant was expressed efficiently (lanes 3–5) and the amplification product matched that produced by expression vector itself (lane 1). RNA from un-transfected cells (U, lane 2) produced a faint band likely representing endogenous Act A transcripts. C, conditioned media samples from each cell population above were incubated with TALON resin or heparin agarose (30 min, 4C). After rinsing, bound proteins were eluted and analyzed by immunoblot with anti-His tag antibody. Note the presence of a thick protein band of ~ 40 kDa (exhibiting the predicted size of the pro-region) in samples from both X2 or X3 pro-region-expressing cells (lanes 2 and 3) but not in samples from un-transfected cells (U, lane 1). Note also that only the X2 encoded pro-region binds heparin agarose (lane 5), while X3 encoded pro-region does not (lane 6).

central cationic surface within each dimer, likely as a result of their location at the C-terminus of each of the two closely juxtaposed mature proteins [31]. These observations and considerations highlight the fact that the HBDs in different proteins differ in location, organization, specific amino acid sequences and number of basic clusters, though they all share fundamental traits of CW motifs. The significant structural variability amongst HBDs in different proteins are likely to have physiological significance in terms of protein availability, distribution and orientation within the extracellular matrix or cell surface as well as with respect to protein interactions with specific HS-proteoglycans and even endogenous protein competitors and antagonists [20]. This conclusion is also sustained by the fact that the domains in many such proteins are evolutionarily conserved [20], but details about their functional significance still need to be fully clarified and understood.

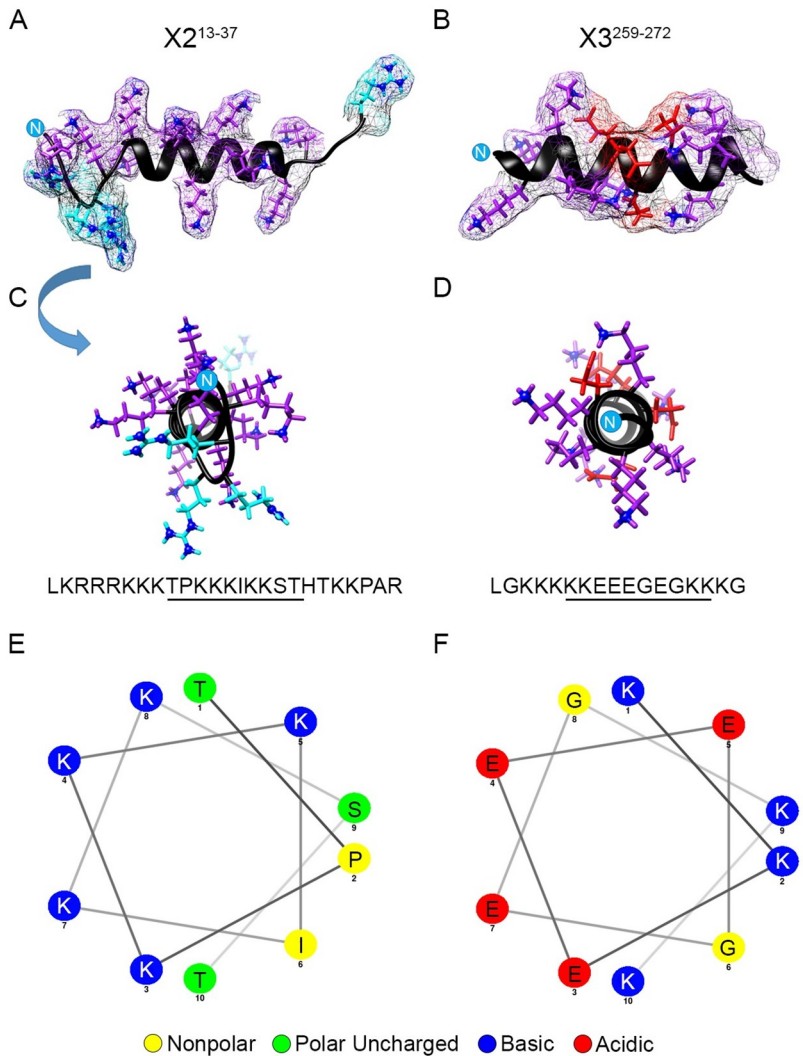

LKRRKKKTPKKKIKKSTHTKKPAR

LGKKKKKEEEGEGKKKG

○ Nonpolar    ● Polar Uncharged    ● Basic    ● Acidic

**Fig 9. Peptides display a helical configuration.** Three dimensional analyses and structure predictions were carried out using the I-TASSER server. A-B, side views of peptides oriented with the N-terminus labeled (N) on the left indicate that both the Act A-X213-37 and Act A-X3259-272 peptides are predicted to possess a helical configuration. C-D, side-view of models depicted in A-B are rotated 90° and provide a frontal view down the axis of the alpha helix, beginning with the N-terminus. Acidic and basic residues are highlighted as follows: Glu; red; Lys, purple; and Arg, cyan. Overall amino acid sequences are noted below, with putative CW motifs underlined. Note in C that the distribution of Lys and Arg residues in Act A-X213-37 elicits uniformly basic surfaces amenable to HS binding. In contrast, the 4 Glu residues in Act A-X3259-272 would disrupt such HS interactions (D). E-F, helical wheel diagrams illustrate the stereological position of amino acids in sequences underlined in C and D, respectively. It is evident that the glutamic acid residues (in red) prevent formation of a uniform basic surface around the helix.

In the standard binding assays used here, peptide X3$^{259-272}$ spanning the previously reported HS-binding region of the X3 variant [19] failed to interact with substrate-bound HS or the cell surface (Figs 6 and 7). Moreover, the entire pro-region of X3 failed to interact with heparin, while the pro-region of the X2 variant did interact (Fig 8). These results are not surprising given that the clusters of basic residue clusters within X3$^{259-272}$ contain 4 intervening glutamic acid residues, which would be predicted to prevent the formation of a uniform cationic HS-binding surface, as our 3D data indicate. How then can our data be reconciled with those of Li et al. [19]? A major difference is that in our solid phase and FACs assays

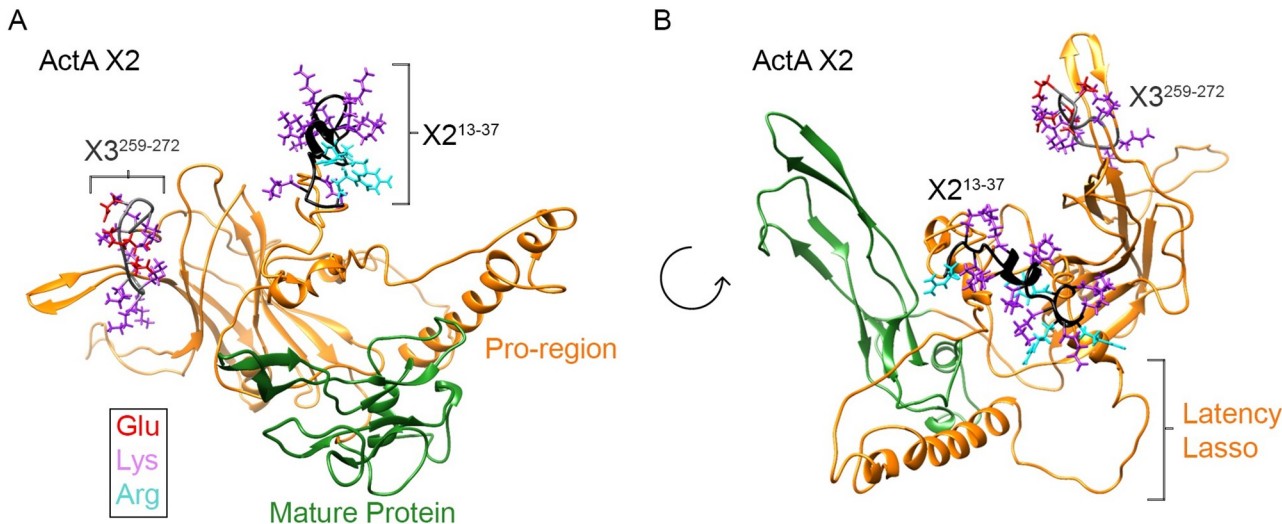

**Fig 10. Predicted structure of Act A variants and location of putative HBDs.** Analyses were carried out with the aid of the I-TASSER server for protein structure and function prediction and visualized using Chimera. Models are based on the crystal structure of pro-TGF-β1 (PDB: 3RJR). *A*, ribbon structure of full length Act A X2 variant monomer in which the pro-region is in orange and the mature protein is in green. Note the location of the high affinity X2$^{13-37}$ HBD (bracketed) and that of the previously reported X3$^{258-272}$ segment (bracketed). Note that the former appears to be exposed to the surface of the monomer that would favor interactions with HS. *B*, same ribbon structure, but viewed after a 90˚ rotation to provide further stereological insights. Also see S1 and S2 Videos in Supporting Information.

(Figs 6 and 7) we made use of peptides corresponding to specific protein domains, while Li et al. made use of full length recombinant X3 variant and perlecan in their solid phase and pull-down experiments. It is possible that full length proteins are able to establish a number of diverse macromolecular interactions with each other in addition to those involving specific segments or motifs interacting with HS chains [44]. HS-rich glypican family members have been shown to interact with Wnts, hedgehogs and other signaling proteins in complex manners, also involving their core protein [45, 46]. In addition, differences in experimental conditions including salt concentration, stringency, washing conditions, buffers, detergents and other parameters could influence outcomes in different studies. We have tested HS derived from different sources including kidney, basement membranes of Engelbreth-Holm-Swarm mouse sarcoma and obtained from different vendors, but the results were the same. Nonetheless, the inhibition of X3 variant/perlecan interactions by heparitinase pre-treatment or mutation of basic residue clusters observed by Li et al. [19] does suggest that those clusters could have some binding roles, despite the presence of acidic residues. However, their affinity and binding strengths for HS could be far lower than those of the new HBD, X2$^{13-37}$, described here. It is worth recalling that the three major follistatin variants differ in their ability to bind HS, with the shorter variant Fs288 exhibiting the highest affinity and the longest variant Fs315 exhibiting the lowest, likely due to the presence of acidic residues [47]. Hence, future studies will be needed to compare side-by-side the binding affinities and kinetics of the X1, X2 and X3 variants and their respective mutants to HS-rich proteoglycans, define the binding hierarchy of different domains, and decipher their roles and underlying mechanisms.

The biosynthesis and elongation of HS chains at serine residues on the proteoglycan core proteins are complex processes involving over 25 enzymes. As indicated above, these step-wise biosynthetic steps results in the creation of segments with specific sulfation and sugar structure patterns, also referred to as 'sulfation codes' [26]. Following secretion, the chains can be further modified by tissue-specific sulfatases that selectively remove sulfate groups at specific positions. The resulting diversity and complexity of sulfated segments along the HS chains are

thought to serve as specific sites of protein interactions [20, 21]. Complementarily, the diverse HBDs identified in different proteins likely add specificity and enable each protein to recognize and bind distinct segments along HS chains with specific sulfation codes. Some of the most characterized protein-HS interactions are those by members of the fibroblast growth factor (FGF) family of signaling molecules [48, 49]. FGFs contain a nine amino acid domain or 'glycine box' that mediates the ability of full-length proteins to selectively bind octasaccharides with different sulfation patterns [48]. Our group recently demonstrated that the HBD present in BMP2 and BMP4 competitively inhibits binding of full-length BMP2 protein to HS, while the HBD in BMP5 does not [31]. While we do not know the preferred sulfation code recognized by ActAX2[13-37] HBD, it may enable the pro-region/mature protein complex for X1 and X2 variants to specifically interact with HS-rich proteoglycans on the cell surface and extracellular matrix, thereby regulating signaling, turn over and function.

As pointed out above, past and recent studies on Act A biology and function have made use of the X3 variant or the mature protein but to our knowledge, the X1 or X2 variants have not been tested. For example, Harrington et al. used recombinant Act A mature protein to clarify the structural basis by which Fs inhibits Act A signaling [14]. They found that Fs dimers wrap themselves around the back of activin dimer wings reorienting their own EGF- and Kazal-like subdomains. Wang et al. carried out a detailed analysis of mechanisms involved in initiation of Act A biological activity, using the X3 variant [18]. They found that intracellular furin cleavage is required to turn on the biological activity of the newly-synthesized X3 variant; as already pointed out above, the signaling potency of the furin-cleaved X3 complex, containing the pro-region and active ligand, was nerly identical to that of mature protein. Notably, this group found that removal of residues 259 to 272, encompassing the X3[259-272] domain described by Li et al. [19], had no effect on biological activity, pointing to a minor if any role of that domain as our data also suggest. Quaternary crystal structure analysis indicated that the two X3 chains form a cross-armed, domain-swapped dimer in which the two central disulfide-linked mature domains wrap around each other and create a connecting bridge to the reminder of the outward oriented globular pro-regions, a configuration different from that in other superfamily members such as TGF-β1. Lastly, Johnson et al. compared the biological activity and half-life of recombinant (cleaved) X3 variant versus mature protein after intravenous injection in male rats [50]. They found that both proteins elicited similar biological activity (monitored by follicle stimulating hormone levels), but the circulating half-life of the entire X3 variant was approximately twice as long as the mature protein ($t_{1/2}$ fast = 12.5 min vs 5.5 min, respectively). Undoubtedly, the above studies and related efforts [5] have provided numerous important and novel insights into Act A biology. However, in view of the data described here, one has to wonder whether testing the X3 variant and mature protein only can actually uncover and clarify the overall physiologic spectrum of Act A activities and underlying mechanisms. Given that the X1 and X2 variants have a more extended N-terminal pro-region and possess a high affinity HBD that is absent in the X3 variant, the X1 and X2 variants could behave and act differently than the X3 variant or mature protein. For instance, the half-life of systemically-injected X1 and X2 variants is likely to be significantly different compared with that of X3 variant or mature protein, given their expected ability to interact with multiple constituents in both blood and tissues through their high affinity HBD. Interactions of X1 and X2 variants with follistatin could also differ from those of X3 due to their larger structure and presence of the high affinity HBD (X2[13-37]) which could affect the binding of Fs288 or Fs315 variants. Within developing embryonic tissues, Act A is thought to form signaling gradients, but this important process has mainly been studied using exogenous mature protein [51]. It will be interesting to reexamine these issues using the X1 and X2 variants and determine endogenous Act A variant expression, diffusion and gradient formation. In sum, it is possible that previous observations

and conclusions derived from studies utilizing mature Act A or the X3 variant may need to be verified and extended by also focusing on the X1 and X2 variants.

In the musculoskeletal research field, interest in Act A biology has been re-invigorated by recent evidence of its pathogenic roles in the pediatric congenital disorder Fibrodysplasia Ossificans Progressiva (FOP) [52]. FOP is characterized by episodic formation of extraskeletal cartilage and bone tissue–a process collectively termed heterotopic ossification (HO)—at multiple sites, eventually involving much of the body [53]. HO is preceded by local inflammation and over time, the accumulated ectopic tissue causes multiple health problems and impediments—including skeletal deformities, chronic pain, growth impairment and joint ankylosis—that increasingly interfere with basic daily functions and can lead to breathing difficulties and premature death. FOP patients bear a missense activating mutation in *ACVR1* that encodes the BMP receptor ALK2 [54]. It was previously thought that activity of mutant ALK2 would be boosted by BMP ligands present at HO initiation sites; given that ALK2-driven BMP signaling mediated by canonical phosphorylated SMAD1/5/8 is chondrogenic, its enhanced activity would lead to local chondrogenesis followed by osteogenesis and ultimately, HO development [55]. However, recent studies have indicated that unlike wild type ALK2, mutant ALK2 can also respond to Act A and elicits pSMAD1/5/8 signaling, thus directly linking inflammation to ectopic chondrogenesis and HO formation [52]. These interesting and important studies have left many questions unanswered. It is assumed that inflammatory cells invariably recruited to the prospective site of HO formation are the source of Act A, but to date, this has not been demonstrated. Further, it is unclear which Act A variant(s) are expressed and how the mature protein ligand becomes available for interaction with mutant ALK2 in skeletogenic cells. Likewise, Act A normally signals via pSMAD2/3 [2], and it remains unknown how its interactions with mutant ALK2 would elicit a switch to pSMAD1/5/8 signaling and what combination(s) of type I and type II receptors are involved. Given our demonstration of the high affinity HBD in the pro-region of X1 and X2 Act A variants, it will be interesting to consider the possibility that synthetic mimetic HS oligomers [56] could be used to sequester Act A precursors, rendering the mature protein less available for action at sites of HO formation.

## Experimental procedures

### Reagents

Human Placental (AM7950), Liver (AM7960) and Lung (AM7968) RNA, NeutrAvidin (NA), NA-HRP and NA-DyLight 488 (NA-488) and Fisher exACTGene DNA Ladder (100–10,000 bp) were obtained from Thermo-Fisher. Heparin was obtained from Sagent Pharmaceuticals and HS was obtained from Millipore-Sigma (www.sigmaaldrich.com). Phorbol 12-myristate 13-acetate (PMA; aka 12-O-Tetradecanoylphorbol-13-Acetate) was obtained from Cell Signaling (#4174). All DNA Oligonucleotide primers were obtained from Integrated DNA Technologies (www.idtdna.com) and are listed in Table 4.

### RNA, PCR & DNA sequencing

Total RNA was isolated using TRIzol Reagent (Thermo-Fisher) following the manufacturers protocol. Five µg of glycogen (Thermo-Fisher AM9510) was added to the aqueous phase prior to the addition of isopropanol to facilitate RNA precipitation. For some experiments, RNA quantification was determined using a Nanodrop spectrophotometer. cDNA was prepared from 2 µg of purified RNA, using oligo dT as primer, supplied with the Verso cDNA synthesis kit (Thermo-Fisher # AB1453A) or SuperScript IV Reverse Transcriptase (Thermo-Fisher #18090010) following the manufactures protocol. All PCR reactions were performed using Act A specific primers and GoTaq Green Master Mix (Promega) following the manufacturers

protocols. PCR reactions were optimized by gradient PCR, using an Applied Biosystems Veriti Thermal Cycler. PCR products were size fractionated on 1% agarose gels, the bands were isolated, DNA was extracted from gel slices using Micro Bio-Spin columns (Bio Rad# 7326204) and sequenced. All DNA sequencing was performed by the NAPCore Facility at CHOP.

## Design of Act A protein expression constructs

The entire open reading frame for Act A variants X1, X2 and X3 were cloned into the EcoRI/ XbaI site of the mammalian expression vector pcDNA3.1 (+). A poly-His tag was added to the 3' end (i.e., C-terminus) to facilitate purification via immobilized metal affinity chromatography (IMAC). The pro-region from each variant was cloned in an analogous manner, except that the His tag was placed before the Furin cleavage site (QSEDHPH<u>HHHHHHH</u>) (see Table 5). Each expression construct was analyzed by Sanger DNA sequencing for verification. For transfection, 2.5 µg of plasmid DNA plus 7.5 µl FuGENE 6 Transfection Reagent (Promega) was added to 0.25 ml Opti-MEM medium and incubated for 15 minutes at room temperature. The transfection mix was subsequently added to cells growing in medium containing 10% fetal calf serum and 24 hr later, the medium was changed to Serum-free media containing Insulin-Transferrin-Sodium Selenite supplement (Roche 11074547001) and G418 (100 µg/ml). Cells were grown for 3 days, the medium was collected, clarified by centrifugation and incubated with TALON Metal Affinity Resin (Takara Bio USA). The resin was washed with PBS, transferred to a Micro Bio-Spin 6 Column (Bio-Rad); bound protein was eluted with 0.5M Imidazole. Purified proteins were analyzed on SDS-polyacrylamide gels/Western blots under reducing conditions and probed with anti-His Tag Monoclonal Antibody (HIS.H8). Western blots were developed using HRP- or IRDye- tagged Secondary Antibodies (Li-Cor).

## mRNA expression of Act A constructs

To evaluate expression of Act A expression constructs, transfected cells were washed and pelleted. RNA was isolated by combining Trizol and Direct-zol RNA MiniPrep Kit (Zymo Research #2050) following the manufacturers protocols. During isolation, RNA is treated with DNAase, to remove any contaminating plasmid or genomic DNA. RNA concentration and purity was assessed using a NanoDrop 2000 spectrophotometer. Act A mRNA expression was evaluated using forward and reverse primers by RT-PCR.

## Biological activity of secreted Act A

To assess biological activity of the expressed Act A variants, THP1 cells were incubated with 3 ml of conditioned medium obtained from cells transfected with each Act A variant for 2 hr, in the absence or presences of an ActA neutralizing antibody (BioLegend # 94746) the cells were

**Table 5. Human activin A full-length and pro-regions.**

| Variant[1] | Accession[2] | Full-length[3] | Pro-region[3] |
|---|---|---|---|
| INHBA X1 | XM_017012174 | 1774–3417 | 1774–3070 |
| INHBA X2 | XM_017012175 | 1653–3101 | 1653–2754 |
| INHBA X3 | XM_017012176 | 2008–3285 | 2008–2938 |

[1]. Activin A Variant

[2]. NCBI Nucleotide Accession number

[3]. Nucleotides in Full-length or Pro-region expression constructs. Mature ligand is 348 nucleotides long (116 amino acids) at the 3' end of the mRNA.

**Table 6. Activin peptides.**

| Peptide [1] | Sequence[2] | pI[3] |
|---|---|---|
| Act A-X2 13–37 | [B]–GGGLKRRRKKKTPKKKIKKSTHTKKPAR | 13 |
| Act A-X3 259–272 | [B]–GGGGKKKKKEEEGEGKKK | 10.3 |

[1]. Activin Peptide

[2]. Peptide Sequence. Three Gly residues and biotin [B] were added to the N-terminus of each peptide.

[3]. pI: Isoelectric point

pelleted and total RNA was isolated using TRIzol Reagent as described above. Induction of Kruppel-like factor 10 (KLF10, TIEG; accession NM_005655) mRNA, an early response gene induced by Act A and TGF-β was assessed by qPCR.

## Peptides

Peptides encompassing the heparin binding domains, contained the putative Cardin-Weintraub (CW) motif and a minimum of 2 flanking amino acids on the N- and C- terminal sides of the domain. Three Glycine residues were added to the N-terminus of each peptide to provide a flexible linker between an N-terminal biotin tag and the peptide itself (See Table 6). Peptides were purified by reverse phase HPLC and peptide mass was confirmed by MALDI-TOF mass spectrometry. All peptides were synthesized and purified by Peptide 2 (Chantilly, VA). Where indicated, biotinylated peptides were oligomerized into tetramers by incubation with NAHRP (each avidin molecule binding 4 biotins ([57]) as described [31].

## Cells

U937 (human pro-monocytic) and THP-1 (human monocytic) cell lines were grown in RPMI containing 10% fetal bovine serum in a humidified atmosphere of 5% CO2. Cells were treated with PMA (100 ng/ml; 160 nM) for 4 hr (23) or GM-CSF (1 ng/ml) for 8 hr in RPMI containing 1% fetal bovine serum.

## FACS analysis

Expression of CD116 (α subunit of GM-CSF receptor). U937 and THP1 cells were washed with PBS and fixed with 2% buffered formalin on ice for 20 min. The cells were washed with PBS and blocked by incubation in PBS, 1% BSA for 20 min. Cells were incubated with anti-Human CD116-PE (BD 551373) or IgG-κ-PE isotype control (BD 554680) on ice for 2 hr, washed with PBS and analyzed on a BD Accuri Flow Cytometer (FL2 channel), located in the Flow Cytometry Core Laboratory at CHOP.

## Peptide binding to intact cells

U937 cells were washed with PBS and fixed with 2% buffered formalin for 20 min on ice. The cells were washed with PBS and blocked by incubation in PBS, 1% BSA for 20 min on ice. Approximately 106 cells (100 μl) were incubated with peptide tetramers consisting of biotinylated peptide complexed with NA-488. Following incubation for 2 hr on ice, the cells were washed and analyzed on a BD Accuri Flow Cytometer (FL1 channel), located in the Flow Cytometry Core Laboratory at CHOP.

## In Silico modeling

Amino acid sequence alignments were performed using the T-coffee alignment tool (tcoffee. crg.cat). Helical wheel projections were constructed using using NetWheels: Peptides Helical Wheel and Net projections maker (lbqp.unb.br/NetWheels). Secondary structure predictions were carried out using the I-TASSER server for protein structure and function prediction [43] and the resulting structures were visualized using Chimera (www.cgl.ucsf.edu/chimera). The structures of Act A-X1 and Act A-X2 are based on the crystal structure of pro-TGF-β1 (PDB: 3RJR).

## Solid phase binding assays

Nunc MaxiSorp flat bottom 96 well plates were coated with HS in 50 mM carbonate buffer (pH 9.4) overnight at 4˚C as described [31]. All binding assays were carried out in 1 x PBS, 0.1% tween 20 (PBST) containing 1% bovine serum albumin. Plates were incubated with peptides for a minimum of 2 hr at room temperature with gentle shaking. At the termination of the assay, the plates were washed 3 times with PBST and a final rinse with PBS. Plates were developed by addition of HRP substrate O-phenylenediamine dihydrochloride (OPD) in Citrate-Phosphate buffer (25 mM citric acid, 50 mM sodium phosphate, pH 5) and read at 450 nm in a Bio-Tek Synergy HT plate reader. Very low levels of background binding were exhibited by NA-HRP alone or when complexed with biotinylated-BSA.

## Statistical analysis

Statistical evaluation of data was performed using Student's t-test employing GraphPad Prism software (www.graphpad.com).

## Supporting information

**S1 Fig. Genomic organization of human Activin B, C & E.** Each Activin has 2 exons. For each Activin, the top diagram delineates Genomic organization and the lower diagram, spliced mRNA. The chromosomal location and sequence homology of Activins C and E suggest they arose from genome duplication. Exon organization was determined using the BLAT search engine, UCSC Genome Browser (genome.ucsc.edu) and are not drawn to scale.
(DOCX)

**S2 Fig. Protein Sequences of full-length human Activins.** A. Schematic representation of Activin proteins. Each protein consists of an N-terminal signal peptide (SP), a prodomain and mature ligand. Red Arrow: furin cleavage site. B. Protein sequences of human Act A, B, C and E are listed. Exons are shaded in different colors. Underlined sequences are putative heparin binding domains (Act AX[178-102] and Act AX3[259-272]). Residues highlighted in Red have codons shared by adjoining exons. Numbers in parenthesis denote position of the processed, active ligand (**Bold** type).
(DOCX)

**S3 Fig. Border region between prodomain (light gray) and active ligand (dark gray) of Activins.** Proteolytic processing is catalyzed by a proprotein convertase/furin protease between a conserved cluster of basic residues (Arg) and hydrophobic/neutral residue (Gly or Thr; red arrow).
(DOCX)

**S4 Fig. Alignment and structure of human Activin ligands.** A. Amino acid alignments were generated with T-Coffee alignment tool (www.tcoffee.org) and UGENE Integrated

Bioinformatics Tools (ugene.net). B. Models were generated by the I-TASSER server for protein structure and function prediction [Yang, et al. (2015) The I-TASSER Suite: protein structure and function prediction. Nature methods **12**, 7–8] and are derived from PDB: 2ARV. N-terminus is in red.
(DOCX)

**S5 Fig. Sanger sequencing results.** Activin transcripts were amplified from cDNAs prepared from human cells and tissues, using PCR primers listed in Table 4 in the manuscript. PCR reactions were run on 1% agarose gels and products were purified and subjected to Sanger Sequencing. The sequencing results were Blasted (blast.ncbi.nlm.nih.gov/Blast.cgi) to confirm identity. Screen shots of Nucleotide Blast results for human Act A, B, C and E are included. Diagrams below each screen shot show the target sequence and primers used for sequencing (Table 4). Also see Fig 1 in the manuscript and S1 Fig.
(DOCX)

**S1 Video. ActA X1 mp4.** Reconstruction of full-length human ActA X1.
(MP4)

**S2 Video. ActA X2 mp4.** Reconstruction of full-length human ActA X2.
(MP4)

**S1 Raw Images.**
(PDF)

## Acknowledgments

We thank Harlan N. Bradford, Dr. Sriram Krishnaswamy and Dr. Rajan M. Thomas for helpful discussion and assistance with data analysis and Anne Lehman and Dr. Martin Carroll for providing U937 and THP-1 cells. We also thank the Nucleic Acid PCR Research Core (NAP-Core) Facility and Flow Cytometry Core Laboratory at CHOP for invaluable assistance.

## Author Contributions

**Conceptualization:** Maurizio Pacifici, Paul C. Billings.

**Data curation:** Evan Yang, Christina Mundy, Eric F. Rappaport, Paul C. Billings.

**Formal analysis:** Eric F. Rappaport, Maurizio Pacifici, Paul C. Billings.

**Funding acquisition:** Maurizio Pacifici.

**Investigation:** Christina Mundy, Eric F. Rappaport, Paul C. Billings.

**Methodology:** Eric F. Rappaport.

**Project administration:** Maurizio Pacifici.

**Software:** Evan Yang.

**Writing – original draft:** Maurizio Pacifici, Paul C. Billings.

**Writing – review & editing:** Evan Yang, Christina Mundy, Eric F. Rappaport, Maurizio Pacifici, Paul C. Billings.

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
