## [Decision Letter · Decision Letter 0]

22 Aug 2019

PONE-D-19-18061

Identification and characterization of a novel heparan sulfate-binding domain in Activin A longest variants and implications for function

PLOS ONE

Dear Dr. Billings,

Thank you for submitting your manuscript to PLOS ONE. After careful consideration, we feel that it has merit but does not fully meet PLOS ONE’s publication criteria as it currently stands. Therefore, we invite you to submit a revised version of the manuscript that addresses the points raised during

We would appreciate receiving your revised manuscript by Oct 06 2019 11:59PM. To enhance the reproducibility of your results, we recommend that if applicable you deposit your laboratory protocols in protocols.io, where a protocol can be assigned its own identifier (DOI) such that it can be cited independently in the future. For instructions see: http://journals.plos.org/plosone/s/submission-guidelines#loc-laboratory-protocols

We look forward to receiving your revised manuscript.

Kind regards,

Nikos K Karamanos, Ph.D.

Academic Editor

PLOS ONE

Journal Requirements:

3. Regarding blot/gel data: PLOS ONE now requires that submissions reporting blots or gels include original, uncropped blot/gel image data as a supplement or in a public repository. This is in addition to complying with our image preparation guidelines described at https://journals.plos.org/plosone/s/figures#loc-blot-and-gel-reporting-requirements. These requirements apply both to the main figures and to cropped blot/gel images included in Supporting Information. If the manuscript is positively reviewed, we will ask the authors to provide any missing raw image data for blot/gel results when they submit their first revision. As part of your review, please ensure that figures reporting blot or gel images comply with the journal’s image preparation guidelines and that the original data are provided following the journal’s request.  If you have any questions or concerns about blot/gel figures or data for this submission, please email us at plosone@plos.org before issuing a decision letter.

Reviewers' comments:

Reviewer's Responses to Questions

**Comments to the Author**

1. Is the manuscript technically sound, and do the data support the conclusions?

Reviewer #1: Yes

Reviewer #2: Yes

2. Has the statistical analysis been performed appropriately and rigorously? 

Reviewer #1: Yes

Reviewer #2: I Don't Know

3. Have the authors made all data underlying the findings in their manuscript fully available?

Reviewer #1: Yes

Reviewer #2: Yes

4. Is the manuscript presented in an intelligible fashion and written in standard English?

Reviewer #1: Yes

Reviewer #2: Yes

5. Review Comments to the Author

Reviewer #1: In this work, authors studied the capacity of the variants X1 and X2 of Activin A to bind heparan sulfate. The experimental procedures included the analysis of human placental RNA and the exposure of human monocytic-like cells to phorbol ester and GM-CSF that stimulates the differentiation to macropohages, where there was noted an increased RNA synthesis of X1 and X2. Synthesis and secretion of all the three variants was tested by transfecting AD293 and COS1 cells, where X2 and X3 were detected. Authors were then showed the binding characteristics of the variants and 3D-structure and protein conformation.

Generally, this manuscript is very well written, clear and well presented. Experiments contain controls and the discussion of the results presents the contribution of this work in the to-date literature.

Reviewer #2: This manuscript is a relevant contribution in Act A studies. Authors successfully showed the expression and biological activity of X1 and X2 variants. Finally, they presented data regarding X1 and X2 interactions with heparan sulfate, describing a possible regulation mechanism of Act A-related processes. The manuscript is clear and well written and, in order to further improve the quality of the paper, please find bellow some minor suggestions.

Abstract - Please specify in what context the other variants are expressed or change the following sentence adequately - “Here, we asked whether the other variants are expressed and what additional structural features may be provided by their longer pro-regions”.

Discussion - I suggest including a topic in discussion associating Act A interactions with HS regarding HS’ sulfation pattern.

Please include statistical information, such as applied tests and p values.

6. PLOS authors have the option to publish the peer review history of their article (what does this mean?). If published, this will include your full peer review and any attached files.

Reviewer #1: No

Reviewer #2: No

---

## [Author Response · Author response to Decision Letter 0]

27 Aug 2019

Aug 27, 2019

RE: PONE-D-19-18061; Identification and characterization of a novel heparan sulfate-binding domain in Activin A longest variants and implications for function

Nikos K Karamanos, Ph.D.

Academic Editor

PLOS ONE

Dear Dr. Karamanos:

Enclosed, please find a revised version of our above referenced manuscript (PONE-D-19-18061). We thank you and the Reviewers for the favorable evaluation of our work and giving us the opportunity to submit a revised manuscript. Below is our response to each comment and question raised by the Editor and Reviewers. Changes to the original version are indicated in Bold type. 

Editorial comments

Response 

As requested, we have modified out manuscript to conform to PLOS ONE's style requirements and file naming. 

Response 

As requested, we have included captions for our Supporting Information files at the end of our manuscript, and updated any in-text citations to match accordingly.

3. Regarding blot/gel data: PLOS ONE now requires that submissions reporting blots or gels include original, uncropped blot/gel image data as a supplement or in a public repository. This is in addition to complying with our image preparation guidelines described at https://journals.plos.org/plosone/s/figures#loc-blot-and-gel-reporting-requirements. These requirements apply both to the main figures and to cropped blot/gel images included in Supporting Information. If the manuscript is positively reviewed, we will ask the authors to provide any missing raw image data for blot/gel results when they submit their first revision. As part of your review, please ensure that figures reporting blot or gel images comply with the journal’s image preparation guidelines and that the original data are provided following the journal’s request. 

Response 

As requested, we have included uncropped blot/gel images. 

Reviewers Comments

Reviewer #1: In this work, authors studied the capacity of the variants X1 and X2 of Activin A to bind heparan sulfate. The experimental procedures included the analysis of human placental RNA and the exposure of human monocytic-like cells to phorbol ester and GM-CSF that stimulates the differentiation to macropohages, where there was noted an increased RNA synthesis of X1 and X2. Synthesis and secretion of all the three variants were tested by transfecting AD293 and COS1 cells, where X2 and X3 were detected. Authors were then showed the binding characteristics of the variants and 3D-structure and protein conformation.

Generally, this manuscript is very well written, clear and well presented. Experiments contain controls and the discussion of the results presents the contribution of this work in the to-date literature.

Response 

We thank Reviewer 1 for their favorable and encouraging assessment of our work. 

Reviewer #2: This manuscript is a relevant contribution in Act A studies. Authors successfully showed the expression and biological activity of X1 and X2 variants. Finally, they presented data regarding X1 and X2 interactions with heparan sulfate, describing a possible regulation mechanism of Act A-related processes. The manuscript is clear and well written and, in order to further improve the quality of the paper, please find bellow some minor suggestions.

Abstract - Please specify in what context the other variants are expressed or change the following sentence adequately - “Here, we asked whether the other variants are expressed and what additional structural features may be provided by their longer pro-regions”.

Response 

As suggested, we have revised our original statement, which now reads: Here, we asked whether these other variants are expressed by human cells in vitro and what additional structural features are contained within their longer pro-regions.

Discussion - I suggest including a topic in discussion associating Act A interactions with HS regarding HS’ sulfation pattern.

Response 

As suggested by the Reviewer, we have added a new paragraph to the Discussion section of the revised manuscript. 

 We hope the revisions and modifications made to our manuscript fully address the Reviewers’ comments and questions and make our study acceptable for publication. We would like to thank once again the Editor and Reviewers’ for their careful evaluation and constructive criticism of our work.

Sincerely, 

Paul Billings, Ph. D.

The Children's Hospital of Philadelphia

Division of Orthopaedic Surgery

Philadelphia, PA 19104

Ph: 267-426-9691/425-2077

Email: BillingsP@email.chop.edu

---

## [Decision Letter · Decision Letter 1]

9 Sep 2019

[EXSCINDED]

Identification and characterization of a novel heparan sulfate-binding domain in Activin A longest variants and implications for function

PONE-D-19-18061R1

Dear Dr. Billings,

We are pleased to inform you that your manuscript has been judged scientifically suitable for publication and will be formally accepted for publication once it complies with all outstanding technical requirements.

With kind regards,

Nikos K Karamanos, Ph.D.

Academic Editor

PLOS ONE

Additional Editor Comments (optional):

Reviewers' comments:

Reviewer's Responses to Questions

**Comments to the Author**

1. If the authors have adequately addressed your comments raised in a previous round of review and you feel that this manuscript is now acceptable for publication, you may indicate that here to bypass the “Comments to the Author” section, enter your conflict of interest statement in the “Confidential to Editor” section, and submit your "Accept" recommendation.

Reviewer #1: All comments have been addressed

Reviewer #2: All comments have been addressed

2. Is the manuscript technically sound, and do the data support the conclusions?

Reviewer #1: Yes

Reviewer #2: Yes

3. Has the statistical analysis been performed appropriately and rigorously? 

Reviewer #1: Yes

Reviewer #2: Yes

4. Have the authors made all data underlying the findings in their manuscript fully available?

Reviewer #1: Yes

Reviewer #2: Yes

5. Is the manuscript presented in an intelligible fashion and written in standard English?

Reviewer #1: Yes

Reviewer #2: Yes

6. Review Comments to the Author

Reviewer #1: (No Response)

Reviewer #2: Authors have fully addressed concerns raised during the revision process reaching a final version that I consider ready for publication.

7. PLOS authors have the option to publish the peer review history of their article (what does this mean?). If published, this will include your full peer review and any attached files.

Reviewer #1: No

Reviewer #2: No

---

## [Editor Report · Acceptance letter]

12 Sep 2019

PONE-D-19-18061R1 

Identification and characterization of a novel heparan sulfate-binding domain in Activin A longest variants and implications for function 

Dear Dr. Billings:

I am pleased to inform you that your manuscript has been deemed suitable for publication in PLOS ONE. Congratulations! Your manuscript is now with our production department. 

With kind regards,

on behalf of

Prof. Dr. Nikos K Karamanos 

Academic Editor

PLOS ONE